# Framework for quality assessment of whole genome cancer sequences

Justin P. Whalley [1,16], Ivo Buchhalter [2,3,4,5], Esther Rheinbay [6,7], Keiran M. Raine [8], Miranda D. Stobbe [1], Kortine Kleinheinz[2], Johannes Werner [2,17], Sergi Beltran[1], Marta Gut[1], Daniel Hübschmann[2,3,9,18,19], Barbara Hutter[5,18], Dimitri Livitz [6], Marc D. Perry [10], Mara Rosenberg [6,7], Gordon Saksena [6], Jean-Rémi Trotta [1], Roland Eils [11,12], Daniela S. Gerhard[13], Peter J. Campbell [8], Matthias Schlesner [2,14] & Ivo G. Gut [1,15 ✉]

Bringing together cancer genomes from different projects increases power and allows the investigation of pan-cancer, molecular mechanisms. However, working with whole genomes sequenced over several years in different sequencing centres requires a framework to compare the quality of these sequences. We used the Pan-Cancer Analysis of Whole Genomes cohort as a test case to construct such a framework. This cohort contains whole cancer genomes of 2832 donors from 18 sequencing centres. We developed a non-redundant set of five quality control (QC) measurements to establish a star rating system. These QC measures reflect known differences in sequencing protocol and provide a guide to downstream analyses and allow for exclusion of samples of poor quality. We have found that this is an effective framework of quality measures. The implementation of the framework is available at: https://dockstore.org/containers/quay.io/jwerner_dkfz/pancanqc:1.2.2.

[1] CNAG-CRG, Centre for Genomic Regulation (CRG), Barcelona Institute of Science and Technology (BIST), Carrer Baldiri i Reixac 4, 08028 Barcelona, Spain. [2] Division of Theoretical Bioinformatics (B080), German Cancer Research Centre (DKFZ), Heidelberg, Germany. [3] Department for Bioinformatics and Functional Genomics, Institute for Pharmacy and Molecular Biotechnology (IPMB) and BioQuant, Heidelberg University, Heidelberg, Germany. [4] Omics IT and Data Management Core Facility (W610), German Cancer Research Center (DKFZ), Heidelberg, Germany. [5] Division of Applied Bioinformatics (G200), Cancer Research Centre (DKFZ), Heidelberg, Germany. [6] Broad Institute of Harvard and MIT, Cambridge, MA, USA. [7] Massachusetts General Hospital Cancer Center and Department of Pathology, Boston, MA, USA. [8] Wellcome Sanger Institute, Hinxton, UK. [9] Department of Pediatric Immunology, Hematology and Oncology, University Hospital Heidelberg, Heidelberg, Germany. [10] Department of Radiation Oncology, University of California, San Francisco, CA, USA. [11] Center for Digital Health, Berlin Institute of Health (BIH) and Charité - Universitätsmedizin Berlin, Corporate Member of Freie Universität Berlin, Humboldt-Universität zu Berlin, Berlin, Germany. [12] Health Data Science Unit, Heidelberg University Hospital and BioQuant, Im Neuenheimer Feld 267, 69120 Heidelberg, Germany. [13] Office of Cancer Genomics, National Cancer Institute, US National Institutes of Health, Bethesda, MD, USA. [14] Bioinformatics and Omics Data Analytics (B240), German Cancer Research Centre (DKFZ), Heidelberg, Germany. [15] Universitat Pompeu Fabra (UPF), Barcelona, Spain. [16]Present address: Wellcome Centre for Human Genetics, University of Oxford, Roosevelt Drive, Oxford, UK. [17]Present address: Department of Biological Oceanography, Leibniz Institute of Baltic Sea Research, Seestraße 15, Rostock, Germany. [18]Present address: Computational Oncology, Molecular Diagnostics Program, National Center for Tumor diseases (NCT) Heidelberg and German Cancer Research Center (DKFZ), Heidelberg, Germany. [19]Present address: Heidelberg Insititute for Stem cell Technology and Experimental Medicine (HI-STEM), Heidelberg, Germany. ✉email: ivo.gut@cnag.crg.eu

Combining whole-genome sequencing data from individual projects has many advantages: increased statistical power, the ability to extend hypotheses across several projects and the possibility of asking biological questions covering a wider range of phenomena[1,2]. Many new methods for investigating cancers are making use of the increased availability of cancer sequencing data to better train their methods[3]. However, when the sequencing data comes from different centres, sequenced over a period of several years and under different protocols, great care must be taken to ensure that the sequencing data is of comparable quality, to avoid drawing false conclusions.

There are several quality control (QC) methods that are being used for whole-genome and exome sequencing data. The Cancer Genome Atlas (TCGA) marker papers for single tumour types (see refs. [4–6] for examples from 2014 to 2016) all calculate QC measures such as depth of coverage, batch effects and contamination levels, as part of the Firehose analysis infrastructure[7]. Likewise work[8] from the International Cancer Genome Consortium (ICGC) with samples sequenced at three different centres relied on similar QC measures computed by the Picard toolkit[9]. Another tool used in the quality assessment of cancer genomes is qProfiler[10] by providing summary statistics on the sequenced data. Combing sequencing data from different projects, but based on exome sequencing, Lu et al.[2], carried out meta-analysis of exome data available from the TCGA for 12 cancer types. Their inclusion criteria were based on coverage depth and percentage of exome coverage for both the normal and tumour samples. Other cancer studies have also pointed to the importance of the percentage of the genome covered[11,12] as well as error rates for each of the paired reads[13] as QC measures. Although many useful QC measurements are available, to our knowledge there is no consistent framework for which QC measures to use, or how to combine the QC measures, to report on the quality of a whole-genome sequence.

To assemble and test a framework to determine the quality of samples and provide a score, we analysed the 2959 normal-tumour genome pairs from 2832 donors of the Pan-Cancer Analysis of Whole Genomes (PCAWG) project[1]. The PCAWG cohort consists of 48 projects encompassed in ICGC[14] and TCGA[15]. The size of this dataset and the diversity of the samples, representing many different cancers from varied populations, make it a perfect testing ground for a QC framework. Variation in quality is to be expected as there were 18 different sequencing centres involved and sequencing was performed over a 5-year time-span (2009–2014), in which the sequencing methodology was still evolving rapidly. Even though there were inclusion criteria based on the sequencing platform (Illumina) and minimum sequencing depth, the quality of the sequencing data and how samples from different centres compared to each other remained an open question.

Based on the PCAWG data we selected measures covering five important features to assess the quality of cancer genome sequences: mean coverage, evenness of coverage, somatic mutation calling coverage, paired reads mapping to different chromosomes and the ratio of difference in edits between paired reads. Here an edit is a base in the read which is different to the reference genome. These measurements we computed for both the normal and tumour samples. To summarise the five QC measures, we established a star rating system to cover the range of the highest quality normal-tumour genome pairs, passing the thresholds set for each measurement, to those that had many sequencing quality issues.

## Results

**Quality control measures**. Three of the QC measures are linked to different aspects of the coverage of the genomic sequence;

mean coverage, evenness of coverage and somatic mutation calling coverage. The other two measures indicate discrepancies between the paired reads: mapping to different chromosomes and the ratio of edits between the paired reads compared to the reference genome. We summarise these five measures into a star rating, for easy comparison of each of the sample pair's quality across the dataset. All our analyses are based on the aligned sequences from the PCAWG core pipeline[1]. We did not use duplicate reads, reads with a mapping quality of zero and supplementary alignments (reads that map to more than one place in the genome).

**Mean coverage**. When deciding on what depth to sequence normal-tumour genome pairs to, a trade off has to be made between the advantages of having a high coverage and the cost of sequencing. The deeper the cancer genome is sequenced the greater the confidence in calling somatic events (see Alioto et al.[16] for a comparison of somatic mutation calling at depths up to 300× sequence coverage). A precondition for the inclusion of a donor in the PCAWG study was the availability of a whole-genome sequence of the normal and tumour with 25× coverage or greater. However, the projects submitting these genomes had calculated coverage differently. For standardisation we defined mean coverage as the mean number of reads covering each position in the genome, where the base is known, after low quality and duplicate reads were excluded as to not inflate the number of reads. For the PCAWG cohort, we did not consider overlapping paired reads, as the samples sequenced tended to have large insert sizes coupled with small read lengths (Supplementary Fig. 1). However, projects with protocols that could lead to overlapping paired reads, should consider removing these to not inflate coverage statistics.

The minimum coverage for inclusion to PCAWG was 30×. However, being more stringent in measuring mean coverage through removing low quality and duplicate reads for the normal samples, we found 25× to be a better minimum for normal samples. The tumour samples tended to be sequenced to a deeper depth, hence we decided to keep the threshold at 30×. These minimum criteria were not met by 0.4% of the normal samples and 2.2% of the tumour samples (Supplementary Fig. 2). We believe for future projects these thresholds represent a minimum level. However, for projects that aim for deeper sequencing, these thresholds could be increased to better reflect the quality of the samples.

**Evenness of coverage**. To confidently identify germline variants and somatic mutations, an even coverage across the target area[17], in this case the entire genome, is ideal. For this QC measure, we used two methods to determine whether the genome is evenly covered. One method is to calculate the ratio of the median coverage over the mean coverage (MoM). An evenly covered sequence should have a ratio of one, with the mean value the same as the median value, not skewed by very low or high coverage in certain regions. To decide within what range of values a sample should fall to be regarded as evenly covered, we used the first quartile (Q1), the third quartile (Q3) and the interquartile range (IQR): $Q1 - (1.5 \times IQR)$ and $Q3 + (1.5 \times IQR)$ for the values for this measurement. For the PCAWG cohort this corresponds to a range of 0.99–1.06 for a normal sample and the wider range of 0.92–1.09 for the tumour samples (Supplementary Fig. 3). Given these thresholds were calculated on 2959 whole normal-tumour genome pairs (from 2830 donors), we suggest these thresholds could be just as representative for evenness of coverage for whole-genome samples sequenced for other cancer projects. Though it should be noted that it is possible that if the

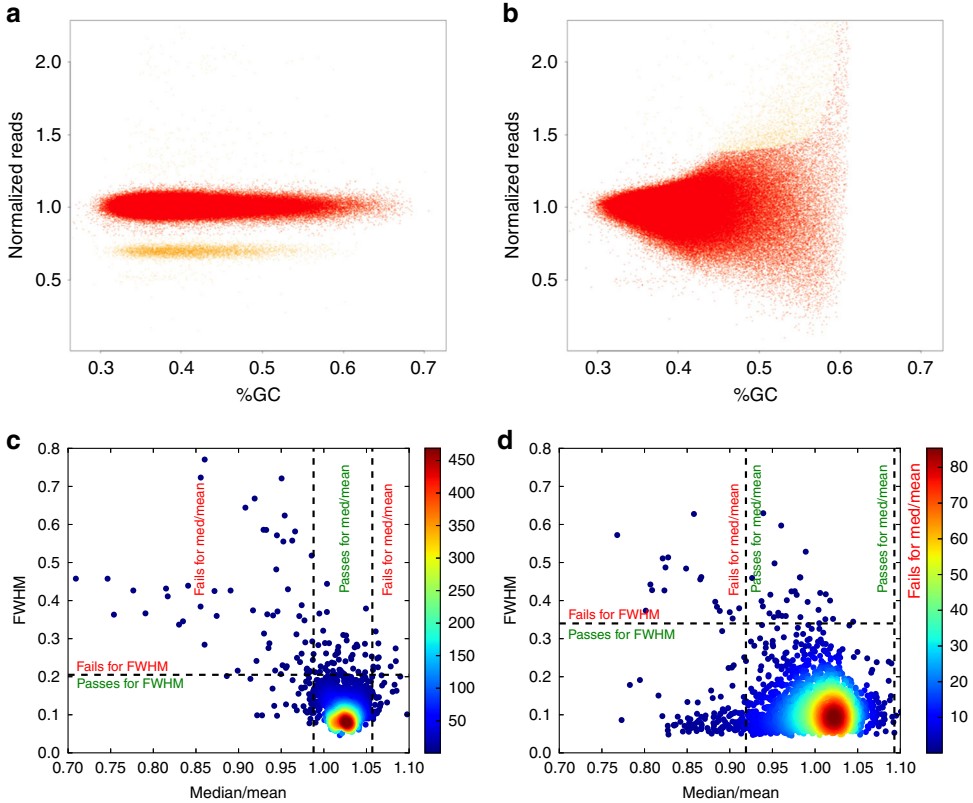

**Fig. 1 Measuring evenness of coverage.** FWHM looks at the GC content versus the normalized coverage for **a** an evenly covered sample and **b** an unevenly covered sample. The main copy number state of the samples, is indicated in red. The yellow cloud represents a different copy number state of a copy number aberrant region. FWHM is calculated on the main copy number state. FWHM is compared to MoM using density scatter plots showing the two measures for **c** normal samples ($n = 2832$) and **d** tumour samples ($n = 2959$). The number of samples overlapping is reflected by the colour at that point as shown by the legend. The dashed lines reflect the thresholds for the evenness measures. These graphs show that while there are certain samples both methods pick out as being unevenly covered, there are also samples picked out by one of the two.

coverage distribution follows a normal distribution, the ratio suggested here will be very close to one regardless of the variance in the coverage, hence the need for more than one measurement for evenness of coverage.

The second measure of evenness looks at the variation of the normalized coverage in ten kilobase genomic windows, after correction for GC-dependent coverage bias (as both GC-rich reads and AT-rich reads are underrepresented in the sequencing results[18]) using the somatic copy number variant (CNV) calling algorithm ACEseq[19]. The main cloud, which corresponds to the main copy number state of the sample, is determined (as shown by the red dots in Fig. 1). The remaining coverage variation is measured as full width at half maximum (FWHM) of the main cloud. This measure is insensitive to copy number aberrations and GC-dependent coverage bias. To determine the thresholds, 1000 whole-genome sequencing samples from different tumour types were used. We chose the thresholds based on clustering of these samples and subsequent visual inspection of the samples that exceeded the threshold to see whether they are valid. Using these results, the thresholds chosen are 0.20 for the normal and the more lenient 0.34 for the tumour, above which the sample would be regarded as having an uneven coverage (Supplementary Fig. 4). These thresholds worked well with the PCAWG project and could be used in other projects.

For MoM and FWHM, there is a greater range of values for the tumour samples than normal samples, potentially due to biologically reasons valid for tumours. For example, large copy number variants could lead to a more unevenly covered sample, distorting the coverage in that region, misleading the evenness of

coverage calculations. However, if the normal sample is unevenly covered, it is more likely due to a sequencing artefact. Hence, we are more stringent for the normal than the tumour samples. The two evenness measures identify different samples as having uneven coverage (Fig. 1). Spearman's rank correlation coefficient for the two measures suggests that these measures are not correlated for the normal ($\rho = 0.24$) and tumour ($\rho = -0.06$) samples. FWHM is insensitive to GC bias, as the CNV caller corrects for this, while MoM identifies other evenness outliers.

The samples need to be in the respective ranges of the MoM and below the thresholds for FWHM for the normal and the tumour to pass the evenness quality measure, of which is not the case for 6.3% and 5.8%, respectively of the PCAWG samples.

**Somatic mutation calling coverage.** Having the depth and evenness of coverage measured, our next QC measure looks at the effect of these at each base in the cancer genome (both the normal and the tumour sample). This measure gives a summary of how much of the cancer genome is sufficiently covered to call a somatic mutation event. The somatic mutation caller MuTect[20] calculates for each base in the genome if it has sufficient coverage in both the normal and tumour sample, that is at least fourteen reads are present in the tumour and eight reads in the matched normal sample. Based on those requirements, we had to establish the number of bases to consider the sample sufficiently covered. Ideally, the threshold should be high enough to penalise the less well-sequenced samples, while not unduly penalising tumour samples that have had large deletions in the genome resulting in

fewer bases to sequence. Taking into account the largest unambiguous mapping for a female donor (so not including the Y chromosome), which would be 2,835,690,481 bases[21], 2.6 gigabases best suits these two needs and should be valid for other whole genome cancer sequences. This results in 6.0% of the normal-tumour pairs in the PCAWG cohort with fewer bases sufficiently covered than this threshold (Supplementary Fig. 4). Though this metric may bear similarity to mean coverage, using Pearson's correlation coefficient, we find that $\rho = 0.17$ for normal samples and $\rho = 0.46$ for tumour samples.

**Paired reads mapping to different chromosomes**. The two reads from a read pair should represent the ends of a contiguous DNA sequence that depending on the insert size should be a given distance apart (for PCAWG between 200 and 800 bases, see Supplementary Fig. 1). Paired reads mapping to different chromosomes can be due to a rearrangement of genome sequence either as natural variation or somatic mutation. In deciding a threshold based on percentage of paired reads mapping to different chromosomes, we should not penalise sequences with biological causes (such as chromothripsis[22], or more generally, interchromosomal rearrangements). However, an excess of reads mapping to different chromosomes points to a technical artefact. Samples with confirmed high levels of rearrangements and chromothripsis, in our experience, do not have more than 1% of paired reads mapping to different chromosomes. Indeed within the PCAWG cohort, there were 94,674 inter-chromosomal somatic structural mutations called for 2428 tumour-normal pairs[23], which suggests an even smaller proportion of reads mapping to different chromosomes for biological reasons in this cohort. To have a large safety margin, we set the threshold to 3%. Samples with a higher percentage may contain technical artefacts, which will also be true of whole genome, cancer sequences in other projects. Of the normal sequences 14.5% exceed the threshold, as do 13.0% tumour sequences (Supplementary Fig. 5). Interestingly, there are more normal samples failing this measure, which cannot be explained by biological processes. A possible explanation may be that for lower quality samples, the PCR amplification used in preparing libraries causes an increase in two fragments of DNA from different parts of the genome being fused together, as has previously been noted[24]. Consequently, this translates to an increase in percentage of paired reads mapping to different chromosomes.

**Ratio of difference in edits between paired reads**. Damage in sequencing runs has been linked to a global imbalance in edits (where the base sequenced is different to the reference) between read 1 and read 2 in paired end sequencing[25]. The ratio of the edits between paired reads, calculated by finding the ratio between the total number of mismatches on read 1 and read 2 across the whole dataset, for a well-sequenced sample should be close to one. We adjudged samples with a two-fold ratio of edits between the paired reads, or greater, as having something gone wrong in the sequencing cycle resulting in lower data quality. Based on this threshold 4.7% and 4.5% normal and tumour samples fail respectively (Supplementary Fig. 6). This two-fold threshold should translate very well to adjudging quality of other whole-genome sequenced cancer samples.

**Choosing of thresholds**. Across the quality measures, we have used several different methods for choosing the thresholds, with a tendency for leniency. For the mean coverage, we chose the minimum we could do, based on the requirements to be part of the PCAWG cohort. While for evenness of coverage, the interquartile range was used for MoM to identify outliers, hence these

thresholds can change with a more consistent dataset. For the other evenness of coverage method, FWHM, the thresholds were set after looking in depth at a subset of the data. In contrast to MoM, these thresholds are constant for this and any other dataset. Somatic mutation calling coverage is limited by how much of the genome we can cover with short reads. As sequencing technologies improve, covering more of the genome, this threshold should be increased to take this into account. For both paired reads mapping to different chromosomes and ratio of difference in edits between paired reads we had to make a value judgement over the best thresholds. For both we erred on the side of caution with higher thresholds to not unnecessarily penalise biological phenomena as opposed to technical artefact from sequencing. Even so, for both we ended up with a higher percentage of normal samples failing than tumour samples, suggesting we are not penalising the mutative properties of tumours (which we are not expecting to see in normal samples), but are picking up on the quality of the sequencing. It is evident that a different study may want to use different thresholds, though for ours it provides coherence across a heterogeneous dataset. Indeed the heterogeneity provided by looking at sequencing for 48 cancer projects from 18 different sequencing centres, provides some robustness to our quality measures and their thresholds for a wide range of cancer, whole-genome sequencing.

**Summary of the five quality measures**. The five quality measures were selected to provide minimal redundancy in flagging quality issues in normal/tumour paired genome sequences. Each measure reflects a facet of sequencing quality that other measures do not. Fig. 2 shows there is some overlap between certain measures, for example, 75 sample pairs are penalised by both having a high percentage paired reads mapping to different chromosomes and

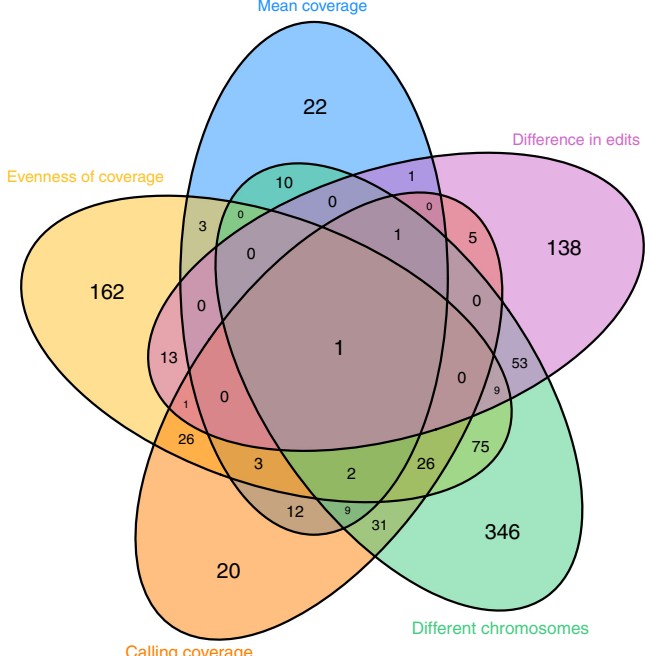

**Fig. 2 Venn diagram showing for which QC measure sample pairs were penalised for.** The outside numbers show the number of sample pairs which fail that QC measure uniquely. The intersections show the number of sample pairs that have failed more than one QC measure. Looking at these overlaps between QC measures, while some measures are closer to each other than others (failing the sample pairs in tandem), they all maintain a large degree of independence.

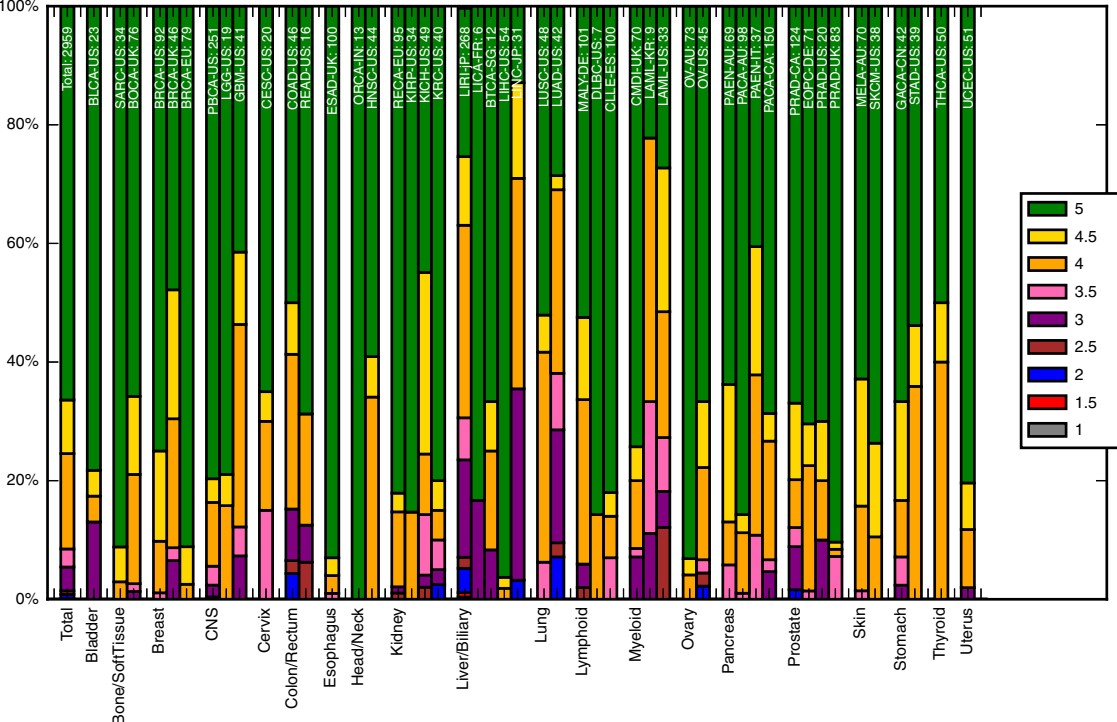

**Fig. 3 Distribution of the star ratings for the PCAWG genomes.** The samples are grouped by tissue type (as labelled along the *x*-axis), and then project. The project name and number of samples in the project are labelled at the top of the bar. The colour of the bar reflects what percentage of samples in the project have that star rating (corresponding to the legend). The bar on the far left shows the results for all samples. The plot demonstrates the varying quality of different projects—differences we believe come from when the genome was sequenced and the sequencing protocol used.

uneven coverage. However, a much higher number of samples is penalised by one of these measures and not the other.

**Star rating system.** We used the five quality measures to construct a star rating for each normal-tumour genome pair. For each QC measure a star is awarded if both the normal and tumour sample pass the threshold. Half a star is awarded if only the normal passes the threshold for the respective QC measures. If only the tumour sample passes and not the normal sample no star is awarded. For somatic mutation calling coverage, a whole star is awarded for passing, none otherwise. The reasons for the extra weighting of the normal sample for the other four measures are that there is no biological reason for not passing the thresholds in the normal sequence and a well-sequenced normal sample is important for calling somatic mutations.

Summing the stars earned for each of the five QC measure results in 66.4% of the normal/tumour sample pairs of the PCAWG being rated as 5 stars. Looking specifically at the different projects (Fig. 3) the quality does not seem to be biased by tissue type (Supplementary Fig. 7) based on detailed molecular subtypes of the tumours in PCAWG[1]. The difference seems to be more at the project level. Unfortunately, there is only limited project metadata on when and which protocol was used to sequence the samples. Detailed metadata was available for 95 donors of the CLLE-ES project[26] (concerning Chronic Lymphocytic Leukaemia), so it could be used as an example. Changes in protocol had an effect on the quality of the sequencing over the 4 years in which CLLE-ES samples were sequenced. For the CLLE-ES project, most notable was the change to a no PCR protocol in 2012, which resulted in improvements to the measures of paired reads mapping to different chromosomes and evenness of coverage. This, in turn, resulted in a measurable change in somatic mutation calling coverage and improvement in star ratings (Supplementary Fig. 8). We found similar results for a

subset of 348 samples sequenced at the Broad Institute (Supplementary Fig. 9), which had metadata recorded in CGHub[27] about the time and instruments used to sequence. We hypothesise that this will be true for other projects as well.

After calculating the star rating for the sequences, we related our QC measures to the calling of somatic single base mutations (SSM), somatic insertion/deletion mutations (SIM) and somatic structural mutations (SStM)[1] in PCAWG (which was done for 2777 samples for SSM and SIMs, and 2712 samples for SStMs). An advantage of using PCAWG is that four callers were used for each set of variants. Looking at the proportion of calls, which all four callers supported, gives us a good idea how the quality of sequencing influences the identification of unambiguous somatic mutations. While the proportion of calls supporting the four callers varies greatly by sample, we find that the samples with four stars or more tended to have higher proportions than samples with less than four stars for SSM, SIM and SStM (with adjusted *p*-values of $3.94 \times 10^{-4}$, $4.86 \times 10^{-4}$ and $1.16 \times 10^{-17}$, respectively, using the Mann–Whitney-*U* test, adjusted using the Bonferroni correction, also see Fig. 4 and Supplementary Table 1).

Taking this analysis further, we used linear regression models to analyse the relation between the proportion of calls supported by four callers and the QC measures (Supplementary Tables 2–4). Though the model only explains a small amount of variance in the data, the results show that an increasing percentage of paired reads mapping to different chromosomes in tumour samples, has a significant negative effect on the proportion of calls supported by four callers for SSM, SIM and SStM. For SSM an increasing mean coverage in tumours has a significant positive effect on the proportion of calls supported by four callers. While for SIM there is a significant negative effect if evenness (as measured by FWHM) decreases in tumours. As for SIM, the unevenness effect is also true in SStM as well as significant negative effects when there is an increase of percentage of paired reads mapping to

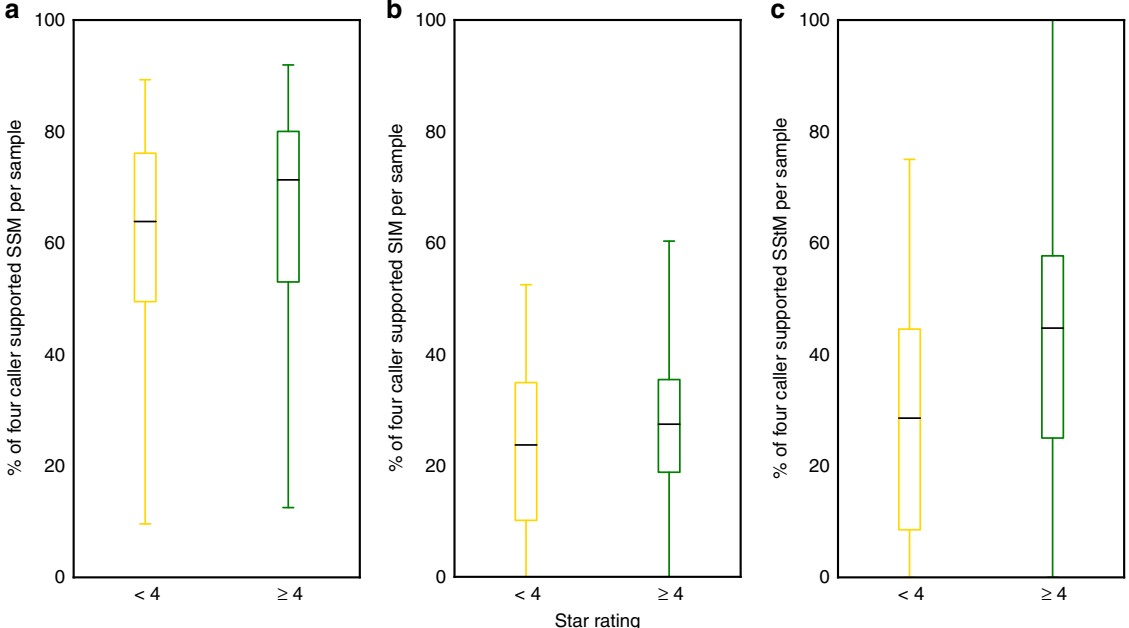

**Fig. 4 Quality of sequencing in samples is linked to consensus in somatic mutation callers.** Box plots comparing percentage of different types of somatic mutations supported by four callers for each sample, with median, the 25% and 75% quartiles shown by the box and the whiskers illustrating 1.5× the interquartile range. Samples with four stars or more tend to have a higher proportion of **a** SSM ($n = 2777$ samples), **b** SIM ($n = 2777$ samples), and **c** SStM ($n = 2712$ samples) supported by four callers than samples with fewer than four stars. These are all significant using a two-sided Mann–Whitney $U$ test, with adjusted $p$-values **a** $3.94 \times 10^{-4}$, **b** $4.86 \times 10^{-4}$ and **c** $1.16 \times 10^{-17}$ (adjusted using the Bonferroni correction, also see Supplementary Table 1).

different chromosomes in normal samples and ratio of difference in edits between paired reads in tumour samples.

The results from this analysis suggest quality of sequencing, measured by our star rating, does have a measurable effect on the downstream analyses. As our QC measures reflect different aspects of sequencing quality, they also have varying levels of importance in using these sequences in the calling of SSM, SIM and SStM.

## Discussion

The established star rating system allows grading the normal and tumour sample sequences by quality in absence of information on how sequencing was carried out, what protocols were used and what problems may have occurred during the sequencing process. The system is not designed to be all encompassing, instead using a small amount of computational resources and time (compared to the actual aligning of the sequences and calling of somatic mutations), we get a good snapshot of the quality of the normal-tumour sample pair sequences on which to call somatic mutations. Likewise having graded the normal-tumour genome pairs with our five-star system, we do not intend researchers to necessarily exclude the lower ranked normal-tumour genome pairs, just to be wary of any conclusions based solely on the lower scoring genomes.

With our star rating system, we sent several samples in PCAWG to the exclusion list due to their poor performance in one of the QC measures. Due to the timing, this did not prevent the downstream analyses being performed. Though anecdotally it would have saved 55 days computational runtime for the one star sample. For all samples that remained, the QC star rating was embedded in the header of the variant call format files. For those projects in PCAWG for which we had metadata, we found that sequencing quality has definitely improved over the time period 2009–2014 in which the samples sequenced. Our results for the CLLE-ES project suggest that in part a protocol change to

PCR-free methods improved sequencing, as in line with best practices from a recent benchmarking exercise[16]. Our data would suggest that when pre-amplification of DNA is needed for whole-genome sequencing, for example in the case DNA is isolated from formalin fixed, paraffin embedded tissue, the star rating system will be helpful when the somatic mutations are interpreted.

Another advantage of our QC is the link to the downstream analyses. In aggregate, sequences with four or more stars have a higher proportion of mutations (SSM, SIM and SStM) identified by all callers. These results suggest overall that higher quality sequence will identify the true positive somatic mutations with higher likelihood.

We believe that our method can be adapted for similar projects that look to use whole-genome sequences from a variety of sources. The thresholds we used based on our experience and applied to this dataset of 2959 normal-tumour genome pairs can also be used to judge the quality of other whole normal-tumour genome pairs. It is worth noting that they represent a trade-off of being severe enough to penalise poor quality while not discriminating against samples with valid biological causes. We also would recommend using our methods to ascertain the quality before downstream analyses. To enable others to use our approach, there is a Docker container, which can be accessed at https://dockstore.org/containers/quay.io/jwerner_dkfz/pancanqc: 1.2.2.

In conclusion, we provide a framework for quality assessment, which opens the door to do large-scale meta-analysis in a more robust setting.

## Methods

**Quality control data.** The individual QC measures and the star rating for each of these normal-tumour sample pairs in PCAWG is provided in Supplementary Data 1: 2959 normal-tumour genome pairs from 2832 donors. Included in our analysis were samples that were later placed on the exclusion or grey list by the PCAWG consortium. Some due quality measures we highlighted, others due to incomplete metadata or other issues like contamination.

**Mean coverage**. The number of reads covering each base of the genome was determined and the mean was calculated:

$$\text{Mean coverage} = \frac{1}{n}\sum_{i=1}^{n} x_i$$

$x_i$ = the number of reads at position $i$.
$n$ = the total number of positions where the base is known.

**Eveness of coverage**. The full width half maximum (FWHM) is estimated during the GC-bias estimation and correction performed by ACEseq[19] (software used to estimate allele-specific copy number from sequencing). ACEseq determines read counts for 10 kb bins and normalizes them for total coverage of the sample. Then a two-step lowess fit procedure is performed to parametrize GC bias and correct for it. An initial lowess curve is fitted to all data points (normalized coverage over the GC content of each 10 kb window) to identify 10 kb windows which belong to the main copy number state (i.e. the copy number state covering the largest fraction of the genome). A second lowess fit using only data points assigned to the main copy number state is then used for parameter assessment and GC bias correction. This two-step fitting procedure prevents influences of copy number aberrations (which might have a different GC content distribution than the whole genome) on the correction function. The coverage values of the 10kb windows are then GC bias corrected by dividing their coverage-normalized read counts by the correction function. The density of the corrected main copy number state coverage is calculated and the FWHM derived from the density curve (Fig. 1).

The median over mean coverage is estimated by using the 10 kb coverage windows used in FWHM, as wells as 261 windows of 5 MB size. The median and mean are calculated and the final result is the median coverage divided by the mean coverage for that sample.

**Somatic mutation calling coverage**. This QC measure was calculated as part of the MuTect mutation calling pipeline[20]. The pipeline counts the number of bases in the genome of the normal sample with at least 8 reads supporting it and the tumour with at least 14 reads supporting it. For our Docker container instead of running the MuTect pipeline, which would be computationally expensive as this will also call SSMs, we use SAMtools[28] for this calculation in the Docker container.

**Paired reads mapping to different chromosomes**. This measures counts the percentage of paired reads that map to different chromosomes. As with the other measurements, reads with a mapping quality of zero, duplicate reads, and supplementary alignments (when reads map to more than one location) were not included. Furthermore, only reads that map to the human autosomes and sex chromosomes are included.

**Ratio of difference in edits between paired reads**. To calculate this ratio of difference in edits between paired reads the following formula was used:

$$\text{Ratio} = \frac{\max\{\sum_{k=1}^{n} r1edits, \sum_{k=1}^{n} r2edits\}}{\min\{\sum_{k=1}^{n} r1edits, \sum_{k=1}^{n} r2edits\}}$$

for $n$ reads where $r1edits$ is the number of edits for read one and $r2edits$ is the number of edits for read two.

The software to calculate this is Bamstats[29] and is included in both our Docker container and the PCAWG Core Pipeline.

**Calculating the star rating**. We used a custom made Python script to determine for each QC measure described above, whether a star, half star or no star should be awarded. The star rating and all the QC measures were saved in tab separated variable file (Supplementary Data 1). The star rating was calculated, and the plots used to illustrate the QC measures of PCAWG here were made, using Python version 2.7.6 and the code uploaded to github: https://github.com/jpwhalley/PCAWG-QC_Graphs.

**Software to calculate the QC measures**. We have collected the custom made scripts and published software together in a Docker container, which can be found at: https://dockstore.org/containers/quay.io/jwerner_dkfz/pancanqc:1.2.2.

The input is an aligned sequence of the normal sample and an aligned sequence of the tumour sample, both files in bam format. In our case we used bams, which can be found in the PCAWG portal: https://dcc.icgc.org/pcawg, all were aligned with bwa-mem 0.7.8-r455 with all alignment scores output and using the default alignment algorithm options against human reference hs37d5.

**Metadata**. The metadata was collected from CGHub for the samples from TCGA, as well as pancancer.info, which has metadata for all samples in PCAWG. Specific metadata concerning the sequencing for the CLLE-ES project was collected internally from the CNAG laboratory information management system. We have included a customised table (Supplementary Data 2) linking the projects to the tumour type. The reason for the customisation is that we have grouped the liver

and biliary samples together, as project LIRI-JP contains samples from both of these tumour types.

**Statistical tests and correlations**. Calculating Spearman's correlation coefficient between the ratio of the MoM coverage and FWHM was done in Python version 2.7.6 (using stats.spearmanr function in the scipy package version 0.18.1).

To test whether the somatic mutation callers were in greater agreement for cancer genome sequences with 4 stars or more we used a two-sided Mann–Whitney-$U$ test and corrected the 3 tests (from SSM, SIM and SStMs) using the Benjamini/Hochberg method with false discovery rate of 0.05 in Python version 2.7.6 (with the stats.mannwhitneyu function and stats.multicomp.multipletests function in the scipy package version 0.18.1) with the results are shown in Supplementary Table 1.

**Linear regression models**. To construct linear regression models to see the relationship between the proportion of calls supported by four callers, we used the lm function of R version 3.3.2. The linear regression model is of the form:

$$Y_i = \beta_0 + \beta_1(X_{i1}) + \cdots + \beta_n(X_{i9}) + \epsilon_i$$

where $i = 1, \cdots, n$ samples for the proportion of calls supported by four callers $Y$. For SSMs and SIMs we had 2657 donors for which we had a normal and tumour sample pair and for which the mutation calls were available (i.e. the samples were not on the exclusion list). For SStMs, there were 2524 samples, as the calling pipeline was not completed on all samples.

$X_{i1,\cdots,9}$ are the nine QC measures (mean coverage for normal and tumour samples, FWHM measures for normal and tumour samples, somatic mutation calling coverage, paired reads mapping to different chromosomes for normal and tumour samples, and the ratio of difference in edits between paired reads for normal and tumour samples) for sample $i$. The median coverage over the mean coverage was not included in the model as this measure is not monotonic increasing or decreasing with respect to the increasing evenness of the coverage.

$\beta_{0,\cdots,n}$ are the coefficients we look to estimate, if they are positive, an increasing QC measure has a positive effect on the proportion of calls supported by four callers, and the reverse if negative.

$\epsilon_i$ represents the errors in the relationship.

All the $p$-values from these linear regression models have been corrected using the Benjamini/Hochberg method with false discovery rate of 0.05.

The results are shown in Supplementary Tables 2–4.

**Reporting summary**. Further information on research design is available in the Nature Research Reporting Summary linked to this article.

## Data availability

The data is from donors involved in PCAWG were recruited by local centres following local protocol including obtaining informed consent. PCAWG was overseen by both the TCGA and ICGC. The whole-genome sequences of the normal and the tumour sample used in this paper can be downloaded from the PCAWG page in the ICGC Data Portal: https://dcc.icgc.org/pcawg. No accession codes are needed, as only data from the PCAWG project is provided on this page in the portal. From the aligned reads downloaded from here, and using our Docker package it will be possible to recalculate our QC measures. We have also included the QC measures and star rating in Supplementary Data 1. Figure 1a, b are plotted by ACESeq[19] during a real time reading of the input coverage files. Source data are provided with this paper.

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

## Acknowledgements

J.P.W., M.D.S., S.B., M.G., J.T. and I.G.G. acknowledge the support of the Spanish Ministry of Science and Innovation through the Instituto de Salud Carlos III and the 2014–2020 Smart Growth Operating Program, to the EMBL partnership and co-financing with the European Regional Development Fund (MINECO/FEDER, BIO2015-71792-P). We also acknowledge the support of the Centro de Excelencia Severo Ochoa, and the Generalitat de Catalunya through the Departament de Salut, Departament d'Empresa i Coneixement and the CERCA Programme. We also acknowledge the support through the European Union's Horizon 2020-funded project EASI-Genomics under grant agreement no 824110. The funding bodies did not have any influence on the design of the study and collection, analysis, and interpretation of data and in writing the paper. The work done by I.B., K.K., J.W., D.H., B.H., R.E. and M.S. was supported by the BMBF-funded Heidelberg centre for Human Bioinformatics (HD-HuB) within the German Network for Bioinformatics Infrastructure (de.NBI) (#031A537A, #031A537C) and the BMBF-funded German ICGC-projects (ICGC-PedBrain: 109252 (German Cancer Aid), 01KU1201A,B; ICGC-MMML: 01KU1002B and ICGC-DE-MINING: 01KU1505E). K.M.R. and P.C. are members of the Cancer Genome Project supported by a Wellcome Trust grant (098051).

## Author contributions

J.P.W., I.B., E.R., K.M.R., K.K., M.D.S. and J.W. wrote the paper, helped develop and apply the methods and analysed the results. S.B., M.G., D.H., B.H., D.L., M.P., M.R., G.S. and J.T. contributed to the development of the methods. R.E., D.S.G., P.C., M.S. and I.G.G. provided project supervision; through feedback and the reviewing of the work done, as well as editing of the paper. I.B. and J.W. constructed the Docker container with code contributions from K.K. and K.M.R.

## Competing interests

The authors declare no competing interests.
