## [Peer Review File · Nature Communications]

Reviewers' comments:

Reviewer #1 (Remarks to the Author): Expert in cancer sequencing

The researchers developed and applied an excellent framework of five quality control measurements to ensure quality assessment of the international PCAWG project from different institutions. Importantly, 66.4% of normal/tumor sample pairs of the PCAWG were rated with 5 stars.

It would be interesting to include a comparison on the validity of final results between 5-star samples (66.4%) and the remaining samples, if available.

Dimitrios H. Roukos MD, PhD
Professor of Cancer Precision Medicine,
Ioannina University, Greece

Reviewer #2 (Remarks to the Author): Expert in cancer bioinformatics

Whalley et al. provide a standardized computational framework for the qualification of normal and cancer whole genome shotgun sequence datasets. The report discusses the importance of the WGSS data quality control, in particular when data to be analyzed was collected by several laboratories over a period of time when sequencing technology was evolving rapidly. The authors propose 5 metrics to assess data quality and provide recommended threshold values to assess the quality and utility of the datasets. An integrated final quality score of tumor-normal data sets pairs is conceptualized as a 'star system', which is effectively an equal weight 'voting' system based on the 5 individual metrics. Utility of the star-based ranking is demonstrated by an increase in edit concordance across 4 variant calling platforms. A github repo is provided with instructions for the installation of a dockerized version of their quality assessment tool.

Synopsis: The report provides a useful and well considered resource for the community, in particular for researchers new to WGSS and it will help to orient the WGSS data quality assessment process more broadly.

Comments:

Level 0 sequencing quality metrics are not discussed: e.g rate of duplicated DNA fragment (PCR or/and optical) and details on duplicated reads, that can indicate issues with the sequencing process; rate of chastity ('vendor quality') failed reads, sequence GC bias, and metrics such as fraction of unmapped pairs or single reads can be indicative of the problem at some point of data generation. Are such metrics uninformative in the context?

Specific comments on suggested metrics:

1. mean coverage is defined as "the mean number of reads covering each position in the genome, after low quality and duplicate reads were excluded as to not inflate the number of reads". In this metric, is the sequence overlap between paired reads taken into account and if so, how. Not considering overlapping reads can introduce an overestimate of coverage values (both globally and locally).

In the mean value calculation, the methods are described as 'the total number of positions where the base is known' - this should be clearly specified.

Lastly, mean coverage threshold values suggested x25 (normal) x30 (tumor); while plausible, it would be useful if authors discuss the rationale behind those threshold values.

One suggestion here can be to select a number of deeply sequenced data sets ($> \times 60$) and to study dependence of variance calling results on the mean coverage by subsampling reads and modeling lower depth of sequencing. One can calculate sensitivity and specificity relative to the full depth data set. Such curve will be very useful and could help to assign significance of the SNV calls in the future experiments as well as FP and FN rate expectations for a data set with a given depth of sequencing.

2. evenness of coverage.

Here two independent metrics are suggested.

The first metric is based on the ratio between mean and median of the coverage distribution and requiring the ratio to be nearly 1 (excluding outliers in the distribution of the ratio across data sets). There appears to be a typo in the formulae provided for the range and it should be read as. $Q1 - 1,5 \times IQR$ and $Q3 + 1,5 \times IQR$ which is a typical expression used to exclude outliers.

While indeed this metric will exclude skewed samples (e.g. with a long tail in the coverage distribution), the effectiveness of the above metric may be questioned. If the coverage distribution follows a normal distribution; then the ratio suggested in the paper will be very close to one regardless of the variance in the coverage.

The second metric calculates a mean coverage in 10Kb bins, taking into account GC bias and then calculate the spread of the mean coverage (taking into account potential effects of copy number variation). However, how GC bias is taken into account is not detailed in the methods.

For both metrics threshold selection justification should be provided.

3. Somatic mutation calling coverage

This metric assesses a fraction of the genome with min coverage a normal sample (≥ 8) and the matching tumor sample (≥ 14) and a threshold 2.6Gb was used.

How strongly is this metric correlated with the coverage metrics?

4. Paired reads mapping to different chromosomes

Not clear how the 3% threshold was selected and why this threshold is the same for both tumor and normal samples (intuitively one would choose a less stringent threshold for the tumor data)

If read pairs aligned to different chromosomes are due to true biological differences, events would be expected to cluster within the specific chromosomal locations (within small region, depending of the mean DNA fragment length of the library). Reads aligning across chromosomes due to technical artifacts would not cluster. Can one apply this property to improve the accuracy of this metric?

5. Ratio of difference in edits between paired reads

Very reasonable to assume that edits (mismatches) should have similar counts for read one and read two. However, an explanation of the statistics used to calculate the ratio (e.g. mean, median) should be provided. The rationale for the selection threshold values should be fully explained.

Minor comments

1. 'Firehose analysis infrastructure' - Reference is missing
2. 'Picard toolkit' - reference is missing
3. BWA-MEM parameters used in alignments are not specified.
4. 'BamStats' - reference is missing
5. Caption to Figure S9: 7th% should be replaced with 75%.

Martin Hirst PhD

Director, Canadian Epigenetics, Environment and Health Research Consortium Network

Associate Professor, Department of Microbiology & Immunology

Associate Director, Michael Smith Laboratories

Michael Smith Laboratories

The University of British Columbia | Vancouver Campus

2185 East Mall | Vancouver BC | V6T 1Z4 Canada

Phone 604 822 6373 | Fax 604 822 2114

Reviewers' comments:

We want to thank the reviewers for their critical reading of the manuscript and their constructive comments. We believe that by addressing their concerns and comments we have managed to clarify issues. The manuscript has improved a lot due to this. Below is a detailed response to all of the comments of the reviewers.

Reviewer #1 (Remarks to the Author): Expert in cancer sequencing

The researchers developed and applied an excellent framework of five quality control measurements to ensure quality assessment of the international PCAWG project from different institutions. Importantly, 66.4% of normal/tumor sample pairs of the PCAWG were rated with 5 stars.

It would be interesting to include a comparison on the validity of final results between 5-star samples (66.4%) and the remaining samples, if available.

Thank you for your kind comments. We agree that the validity of the results comparing the highest quality samples and the remaining samples would be very interesting. Unfortunately this is not available, in part due to our results, the lowest quality samples in PCAWG were disqualified from further analysis in the PCAWG project, as it is likely that any results coming from them would be unreliable, leaving us without a very useful comparator set.

Dimitrios H. Roukos MD, PhD
Professor of Cancer Precision Medicine,
Ioannina University, Greece

Reviewer #2 (Remarks to the Author): Expert in cancer bioinformatics

Whalley et al. provide a standardized computational framework for the qualification of normal and cancer whole genome shotgun sequence datasets. The report discusses the importance of the WGSS data quality control, in particular when data to be analyzed was collected by several laboratories over a period of time when sequencing technology was evolving rapidly. The authors propose 5 metrics to assess data quality and provide recommended threshold values to assess the quality and utility of the datasets. An integrated final quality score of tumor-normal data sets pairs is conceptualized as a 'star system', which is effectively an equal weight 'voting' system based on the 5 individual metrics. Utility of the star-based ranking is demonstrated by an increase in edit concordance across 4 variant calling platforms. A github repo is provided with instructions for the installation of a dockerized version of their quality assessment tool.

Synopsis: The report provides a useful and well considered resource for the community, in particular for researchers new to WGSS and it will help to orient the WGSS data quality assessment process more broadly.

Comments:

Level 0 sequencing quality metrics are not discussed: e.g rate of duplicated DNA fragment (PCR or/and optical) and details on duplicated reads, that can indicate issues with the sequencing process; rate of chastity ('vendor quality') failed reads, sequence GC bias, and metrics such as fraction of unmapped pairs or single reads can be indicative of the problem at some point of data generation. Are such metrics uninformative in the context?

The reviewer makes a very good point. If researchers would have access to these sequencing quality metrics, they should definitely be taken into account, however this is not always the case. For our situation, looking to combine 48 studies, where these metrics are not available, we needed a post-hoc quality control scheme that would allow us to take a decision on whether a dataset was fit for purpose. As we also emphasize in the paper, this cohort was

sequenced across 18 sequencing centres over a period of 5 years. It was deposited in a very heterogeneous manner, with different levels of processing and sequencing metadata. The framework we chose can deal with this heterogeneity and work well as a set of standalone quality control metrics, if needed, or complement the level 0 sequencing quality metrics, if available.

Specific comments on suggested metrics:

1. mean coverage is defined as "the mean number of reads covering each position in the genome, after low quality and duplicate reads were excluded as to not inflate the number of reads". In this metric, is the sequence overlap between paired reads taken into account and if so, how. Not considering overlapping reads can introduce an overestimate of coverage values (both globally and locally).

We agree with the reviewer, if the paired reads overlapped they would inflate our estimation of coverage. In our analysis of the summary statistics from the mapping of the sequencing, the majority of the samples had a large insert size (from 200bp - 500bp) and a small read length (~ 100bp), which makes overlapping paired reads unlikely. We have updated the section on mean coverage and included a supplementary figure (S1) to reflect this analysis, and note for projects using different protocols that they may want to consider this.

In the mean value calculation, the methods are described as 'the total number of positions where the base is known' - this should be clearly specified.

Thank you, we have made this clear in the main part of the manuscript, so it is consistent with what is stated in the methods.

Lastly, mean coverage threshold values suggested x25 (normal) x30 (tumor); while plausible, it would useful if authors discuss the rationale behind those threshold values.

We agree with the reviewer and have changed the final paragraph in this section to emphasize our selection of thresholds. We admit that they are, in part, based on the minimum coverage needed for a sample to be included in PCAWG, with some leniency based on our stricter, but more consistent method of measuring mean coverage. Our concluding sentence remains the same, that this threshold for future projects is dependent on how deep those projects can afford to sequence.

One suggestion here can be to select a number of deeply sequenced data sets (>x60) and to study dependence of variance calling results on the mean coverage by subsampling reads and modeling lower depth of sequencing. One can calculate sensitivity and specificity relative to the full depth data set. Such curve will be very useful and could help to assign significance of the SNV calls in the future experiments as well as FP and FN rate expectations for a data set with a given depth of sequencing.

This is a valid suggestion, and work following on Alioto, T. S. *et al.* [citation #16 in the manuscript], who calculated to a very accurate degree the sensitivity and specificity when sequencing a sample up to 300X with a hand curated gold somatic mutation call set. While with the PCAWG cohort we have the advantage of many more samples, the disadvantage is how somatic mutations were called; by consensus using four callers (as detailed in Campbell, P. J. *et al.* [citation #1 in the manuscript] supplementary methods section 2). While this meant we had very specific calls, this was to the detriment of sensitivity. While this worked well in the context of the PCAWG analysis, it does mean it would be lacking in trying to calculate the sensitivity and specificity relative to sampling depth. Though we agree with the reviewer that future work, reworking this on similar data, using the methods in Alioto, T. S. *et al.*, would be very useful.

2. evenness of coverage.

Here two independent metrics are suggested.

The first metric is based on the ratio between mean and median of the coverage distribution and requiring the ratio to be nearly 1 (excluding outliers in the distribution of the ratio across data sets). There appears to be a typo in the formulae provided for the range and it should be read as. $Q1 - 1,5 \times IQR$ and $Q3 + 1,5 \times IQR$ which is a typical expression used to exclude outliers.

Thank you, we have fixed the typo.

While indeed this metric will exclude skewed samples (e.g. with a long tail in the coverage distribution), the effectiveness of the above metric may be questioned. If the coverage distribution follows a normal distribution; then the ratio suggested in the paper will be very close to one regardless of the variance in the coverage.

We thank the reviewer for pointing this out. We have stated the reviewers point in the manuscript and hence the motivation for a second test for evenness of coverage.

The second metric calculates a mean coverage in 10Kb bins, taking into account GC bias and then calculate the spread of the mean coverage (taking into account potential effects of copy number variation). However, how GC bias is taken into account is not detailed in the methods.

We agree with the reviewer and have reworded and added to the methods to make clear how GC bias is taken into account. As a side, it is worth mentioning that GC bias depends on the sample preparation procedure applied which was not under our control.

For both metrics threshold selection justification should be provided.

We thank the reviewer in highlighting this for this and the other metrics, we have added another subsection to discuss how we chose the thresholds for the metrics.

3. Somatic mutation calling coverage

This metric assesses a fraction of the genome with min coverage a normal sample (≥ 8) and the matching tumor sample (≥ 14) and a threshold 2.6Gb was used.

How strongly is this metric correlated with the coverage metrics?

We have added in the manuscript that the Pearson correlation coefficient for mean coverage and this metric is 0.17 for normal samples and 0.46 for tumour samples. These are sufficiently low, that we believe these two metrics are highlighting different quality issues (though logically they are somewhat related).

4. Paired reads mapping to different chromosomes

Not clear how the 3% threshold was selected and why this threshold is the same for both tumor and normal samples (intuitively one would choose a less stringent threshold for the tumor data)

As we commented above, we have added a thresholds subsection to discuss how we selected the thresholds for the differing metrics. The interesting result of this metric, which we did not expect, is that more normal samples were

penalised at the same threshold than tumour samples. To help explain why we do not see a greater effect in tumour samples, we have added to the manuscript, research from PCAWG on structural somatic mutations [citation #23] that shows a minority of samples in the cohort have structural somatic mutations and they in turn only explain a tiny fraction of paired reads mapping to different chromosomes.

If read pairs aligned to different chromosomes are due to true biological differences, events would be expected to cluster within the specific chromosomal locations (within small region, depending of the mean DNA fragment length of the library). Reads aligning across chromosomes due to technical artifacts would not cluster. Can one apply this property to improve the accuracy of this metric?

We agree with the reviewer that would be very interesting. Looking at the work on PCAWG Structural Variation [citation #23], there are on average ~39 inter-chromosomal structural somatic mutations called for 2,428 tumour-normal pairs (with a maximum of 943 for one pair), called across the cohort, suggesting that the paired reads mapping to different chromosomes are most likely technical or sequencing artifacts and we are very unlikely to penalise biological phenomena with this metric. We have highlighted this in the manuscript.

5. Ratio of difference in edits between paired reads

Very reasonable to assume that edits (mismatches) should have similar counts for read one and read two. However, an explanation of the statistics used to calculate the ratio (e.g. mean, median) should be provided. The rationale for the selection threshold values should be fully explained.

Thank you, we have added to the main manuscript the statistic used to calculate the ratio (in this case it is the total number).

With the very useful comments of the reviewer about thresholds for several QC measures, we have added a subsection on thresholds. In this subsection we discuss the range of thresholds we chose from the minimum for mean coverage, to using the interquartile range for ratio of the median over mean coverage, which could change depending on the consistency of the dataset, to other measures for which we have chosen fixed values. We needed to balance the competing desires of catching poorer sequencing, while not unduly penalising biological phenomena. The reviewer has also discussed methods to possibly fine tune some of these thresholds. We believe that these could definitely be useful, especially for a more homogeneous dataset. However, for our very varied dataset, we have prized robustness of the measures, so they can possibly translate across to other cancer, whole-genome datasets.

Minor comments

1. 'Firehose analysis infrastructure' - Reference is missing

Thank you, the following has been added, following the instructions here

http://gdac.broadinstitute.org/runs/info/DOIs__stddata.html and referenced as a website following the *Nature* formatting guide:

Broad Institute TCGA Genome Data Analysis Center. Firehose stddata 2016 01 28 run.

<https://doi.org/10.7908/C11G0KM9> (2016).

2. 'Picard toolkit' - reference is missing

Thank you, the following has been added, following the instructions here <https://github.com/broadinstitute/picard> and referenced as a website following the *Nature* formatting guide:

Broad Institute, GitHub Repository. Picard toolkit. <http://broadinstitute.github.io/picard/> (2019).

3. BWA-MEM parameters used in alignments are not specified.

Thank you, we have added this information in the Methods section, under the *Software to calculate the QC measures* subsection: bwa-mem 0.7.8-r455 with all alignment scores output and using the default alignment algorithm options against human reference hs37d5.

4. 'BamStats' - reference is missing

Thank you, the following has been added, referenced as a website following the *Nature* formatting guide:

Raine, K. M. et al. Pcap-core. <https://github.com/cancerit/PCAP-core> (2020).

5. Caption to Figure S9: 7th% should be replaced with 75%.

Thank you, this has been fixed.

Martin Hirst PhD

Director, Canadian Epigenetics, Environment and Health Research Consortium Network

Associate Professor, Department of Microbiology & Immunology

Associate Director, Michael Smith Laboratories

Michael Smith Laboratories

The University of British Columbia | Vancouver Campus

2185 East Mall | Vancouver BC | V6T 1Z4 Canada

Phone 604 822 6373 | Fax 604 822 2114

REVIEWERS' COMMENTS:

Reviewer #2 (Remarks to the Author):

I thank the authors for their thoughtful responses to the original suggestions in their revised manuscript. I have no further concerns.